# Effect of Calcium Fortified Foods on Health Outcomes: A Systematic Review and Meta-Analysis

**DOI:** 10.3390/nu13020316

**Published:** 2021-01-22

**Authors:** Gabriela Cormick, Ana Pilar Betran, Iris Beatriz Romero, Maria Sol Cormick, José M. Belizán, Ariel Bardach, Agustín Ciapponi

**Affiliations:** 1Department of Mother and Child Health Research, Institute for Clinical Effectiveness and Health Policy (IECS-CONICET), Ciudad de Buenos Aires 1414, Argentina; belizanj@gmail.com; 2Centro de Investigaciones Epidemiolóicas y Salud Púlica (CIESP-IECS), CONICET, Ciudad de Buenos Aires 1414, Argentina; abardach@iecs.org.ar (A.B.); aciapponi@iecs.org.ar (A.C.); 3Departament de Salud, Universidad Nacional de La Matanza (UNLAM), San Justo 1903, Argentina; romeroiris.b@gmail.com; 4UNDP/UNFPA/UNICEF/WHO/World Bank Special Programme of Research, Development and Research Training in Human Reproduction (HRP), Department of Sexual and Reproductive Health and Research, World Health Organization, 1211 Geneva, Switzerland; betrana@who.int; 5Departament de Diagnóstico por Imágenes, Fleni, Montañeses 2325, Ciudad de Buenos Aires C1428, Argentina; solcormick@hotmail.com; 6Centro Cochrane Argentino-Instituto de Efectividad Clínica y Sanitaria (IECS-CONICET), Ciudad de Buenos Aires 1414, Argentina

**Keywords:** calcium, fortification, systematic review, staple foods, commonly consumed foods

## Abstract

Calcium supplementation and fortification are strategies widely used to prevent adverse outcome in population with low-calcium intake which is highly frequent in low-income settings. We aimed to determine the effectiveness and cost-effectiveness of calcium fortified foods on calcium intake and related health, or economic outcomes. We performed a systematic review and meta-analysis involving participants of any age or gender, drawn from the general population. We searched PubMed, Agricola, EMBASE, CINAHL, Global Health, EconLit, the FAO website and Google until June 2019, without language restrictions. Pair of reviewers independently selected, extracted data and assessed the risk of bias of included studies using Covidence software. Disagreements were resolved by consensus. We performed meta-analyses using RevMan 5.4 and subgroup analyses by study design, age group, and fortification levels. We included 20 studies of which 15 were randomized controlled trials (RCTs), three were non-randomised studies and two were economic evaluations. Most RCTs had high risk of bias on randomization or blinding. Most represented groups were women and children from 1 to 72 months, most common intervention vehicles were milk and bakery products with a fortification levels between 96 and 1200 mg per 100 g of food. Calcium intake increased in the intervention groups between 460 mg (children) and 1200 mg (postmenopausal women). Most marked effects were seen in children. Compared to controls, height increased 0.83 cm (95% CI 0.00; 1.65), plasma parathyroid hormone decreased −1.51 pmol/L, (−2.37; −0.65), urine:calcium creatinine ratio decreased −0.05, (−0.07; −0.03), femoral neck and hip bone mineral density increased 0.02 g/cm^2^ (0.01; 0.04) and 0.03 g/cm^2^ (0.00; 0.06), respectively. The largest cost savings (43%) reported from calcium fortification programs came from prevented hip fractures in older women from Germany. Our study highlights that calcium fortification leads to a higher calcium intake, small benefits in children’s height and bone health and also important evidence gaps for other outcomes and populations that could be solved with high quality experimental or quasi-experimental studies in relevant groups, especially as some evidence of calcium supplementation show controversial results on the bone health benefit on older adults.

## 1. Introduction

Dietary calcium intake in low-income settings is typically low, and around 3.5 billon people are considered to be at risk of calcium deficiency [1,2]. Calcium deficiency leads to osteoporosis, with nearly 9 million fractures annually worldwide, causing people to become bedridden with serious complications [3,4].

Recommendations for calcium intake vary. For individuals over 19 years of age, a daily calcium intake of 1000–1300 mg is recommended [5,6]. The latest recommendations published in 2010 by the US Institute of Medicine (IOM) were established taking into account the bone health needs of healthy individuals [4]. Calcium supplementation can help to address the low calcium intake related problems. The benefits of calcium supplementation seem to be greater in children and adolescents with low calcium intake [7]. Calcium effects on bone health in other age groups are usually evaluated in combination with other micronutrients specially vitamin D, so data for calcium alone are limited [8,9]. The US Preventive Task Force (USPTF) recommends calcium supplementation plus vitamin D based on a 2016 meta-analysis showing a relative risk reduction (RRR) of 15% on the incidence of fractures and a 30% in hip fractures in middle-aged to older adults [7,10]. This is also supported by a study published in 1992 that found that calcium supplementation with vitamin D3 reduces the risk of hip fractures and other nonvertebral fractures among elderly women [11]. On the other hand, as calcium supplements has been related to constipation, bloating and kidney stones, and some evidence suggests they may cause a small increase in the risk of myocardial infarction in elderly adults a recent review does not recommend the use of calcium supplements in healthy community-dwelling adults [12].

However, new evidence has shown that adequate calcium intake also has health benefits beyond bone health [13]. Appropriate calcium intake has been associated with lower blood pressure particularly among young subjects, prevention of hypertensive disorders of pregnancy, colorectal adenomas, reduced low-density lipoprotein (LDL) cholesterol levels and lower blood pressure in the offspring of women taking sufficient calcium during pregnancy [14,15,16,17,18] All of these health outcomes have high burden of disease and should also be considered to set recommendations.

Calcium supplementation compared to placebo has shown a 55% (95% CI: 35 to 69%) of RRR of preeclampsia [17]. In populations with low calcium intake below 800 mg/day, the effect was even higher (RRR 64%). A further RRR of 34% was observed in women with good supplementation adherence to 500 mg of calcium a day starting preconceptionally and up to mid-pregnancy [19].

In 2011, WHO guidelines for the prevention of preeclampsia recommended calcium supplementation with 1.5–2.0 g per day from 20 weeks pregnancy, particularly in a population with low calcium intake [20]. The 2020 update of the guidelines recommend that women achieve an adequate calcium intake through locally available foods and suggest that food fortification might be an appropriate strategy to fulfil this recommendation, as there are major acceptability and feasibility concerns with recommendation to increase calcium intake using supplements [19,21]. Previous studies have highlighted that the pill burden of calcium supplementation is high, and adherence to medications is lower when multiple doses are required [21]. Also, a small proportion of individuals experience side effects such as constipation, and these side effects may be worse at higher doses [22]. Feasibility of implementing this strategy to reach all the population is also a concern as bulk weight of calcium supplements is high resulting in significant logistical cost burdens due to transport, storage and distribution [23,24]. Besides there are economic constrains when supplements need to be paid by the users. These acceptability and feasibility issues are major barriers to scale-up and adoption of calcium supplementation by health systems in low- and middle-income countries [17,25].

For these reasons, the discussion on calcium supplementation has moved in recent years to food fortification as a more effective means to achieve adequate levels of calcium intake and reaching entire populations. Food fortification has been identified as one of the most cost-effective interventions to address micronutrient deficiencies in populations [26]. In addition, fortification would reach populations with less contact with the health systems increasing the benefits beyond pregnancy, and beyond women [13].

In this review and meta-analysis, we aimed assess the effectiveness and cost-effectiveness of commonly consumed food fortified with calcium on calcium intake and on selected clinically relevant health outcomes. This information could help to design a population-based food fortification implementation scheme, as a public health strategy to improve calcium intake.

## 2. Materials and Methods

We performed a systematic review and meta-analysis following Cochrane methods [26], and the PRISMA statements for reporting [27,28]. The protocol is registered in the University of York’s PROSPERO database for systematic reviews (CRD42020150823). 

### 2.1. Eligibility Criteria 

#### 2.1.1. Type of Studies

We considered the following study designs: complete and incomplete economic evaluation or cost studies, randomized controlled trials, controlled before-and-after trials (CBA), uncontrolled before-and-after trials (UBA), interrupted time series (ITS) designs with at least three data points before and after the intervention, with or without comparison groups, and cohort studies. Systematic reviews were considered as a source of studies and other study designs were used only to describe the intervention.

#### 2.1.2. Type of Participants

We included studies involving participants of any age or gender. We excluded those studies exclusively performed in special populations (i.e., only patients with a unique condition). We included populations with any levels of calcium intake and in any country, region or setting.

#### 2.1.3. Type of Fortification (Interventions)

We prioritized the inclusion of interventions of commonly consumed foods fortified exclusively with calcium. However, in their absence, we also included interventions that fortified foods with calcium salts, calcium from milk extracts, or commercially available foods with high calcium that allowed to increase calcium intake of the population. We included interventions where foods had the addition of calcium and/or other nutrients carried such as those minerals from milk (including phosphorous, magnesium or zinc), even if the increase of this other nutrients was only in the intervention group. We excluded those interventions where foods had calcium and vitamin D unless the same increment on vitamin D was also in the control group. We only included studies with calcium and vitamin D fortification when the outcome was dietary calcium intake, since there is no evidence that vitamin D interferes calcium intake. Otherwise, we only considered calcium fortification as an intervention. We included studies that compared the fortified food with similar unfortified food, with a food with lower content of calcium, with usual diet, or with supplement placebo.

#### 2.1.4. Type of Outcomes

The primary outcome was calcium intake reported as mg per day, and health outcomes related to calcium intake such as bone health, bone metabolism, blood pressure, preeclampsia (for pregnant women), cardiovascular outcomes, hypertension, lithiasis, cholesterol, weight, height and body mass index (BMI). We also aimed to characterize the cost, cost-effectiveness and budget impact analyses of commonly consumed food fortified with calcium.

### 2.2. Search Strategy for Identification of Studies and Data Sources

We searched, from inception to 8 June 2019, PubMed, Agricola, EMBASE, CINAHL and Global Health. We did not apply language limitations or publication date restrictions for the search, however we only included studies in English and Spanish. For studies with multiple publications, we used the publication reporting more information. The search was performed on 9 December 2019. Appendix B
Table A1 shows the search strategy and the list of search terms. As for grey literature, we ran a generic Internet search, searched Google Scholar and inspected the Food and Agriculture Organization (FAO) website, and those of Ministries of Health of countries are implementing calcium fortification, such as the UK. For economic studies we searched EconLit, of the American Economic Association and the Cost-Effectiveness Analysis (CEA) and the Registry of the Center for the evaluation of Value and Risk in health, Tufts Medical Center.

### 2.3. Data Collection and Analysis

Two reviewers independently screened titles and abstracts in duplicate. We retrieved the full text of all potentially relevant studies. The full text of these studies were retrieved and read also in duplicate; those that fulfilled the aforementioned selection criteria were included in the review. Disagreements were resolved by consultation with a third reviewer, and agreement on discordant decisions was reached by consensus. We extracted data on study title, author, publication date, country, city, included population, setting, study design, intervention food, fortification level, comparison group and outcomes.

#### 2.3.1. Risk of Bias Assessment

Two reviewers independently performed quality appraisal for each study, and disagreements were resolved by discussion or consultation with a third reviewer. For randomized controlled trials (RCTs), we used the Cochrane risk of bias tool, which assesses selection bias, performance bias, detection bias, attrition bias, and reporting bias rating each component as “high”, “low”, or “unclear” for each risk of bias component. For Non-Randomized studies, we used the Study Quality Assessment Tools for critical appraisal of the internal validity of each type of study (https://www.nhlbi.nih.gov/health-topics/study-quality-assessment-tools). Economic studies were assessed with the CHEERs checklist [29]. Independently selection, data extraction and risk of bias assessment of included studies was performed using the Covidence software [30].

#### 2.3.2. Statistical Analyses

We performed the meta-analysis using RevMan 5.4 [31]. For dichotomous data, we used odds ratios (OR) and risk ratios (RR) with 95% confidence intervals (CI). For continuous data, we used the mean difference (MD) with 95% CI, if outcomes were measured in the same way between trials. We used standardized mean difference (SMD) with 95% CI to combine trials that measured the same outcome but used different methods of measurement. We used the inverse variance method (IVM) to combine before and after studies with experimental designs. We assessed heterogeneity using I^2^ along with a visual inspection of forest plots. 

We planned to conduct the following subgroup analysis: study design (RCTs vs non-randomized trials, quasi experimental studies and observational studies (Non–RCT)), age group (Children and adults), menopausal status, level of fortification (low, moderate, high) and basal calcium intake. We calculated a pooled effect size for each subgroup. Due to the scarcity of data we couldn’t perform the analysis for WHO region and continent subgroups. 

## 3. Results

The search strategy retrieved a total of 3186 and after removing duplicates, 2122 articles remained for title and abstract screening. A total of 1978 studies excluded for title and abstract screening and of the 144 articles screened for full text eligibility. Finally, 20 studies fulfilled the inclusion criteria, 18 for clinical outcomes (15 RCTs, three non-RCT), and two economic evaluations (see Figure 1). The 15 RCTs were included in the meta-analysis and five studies were only described as they we were not able to include them in the meta-analysis. The main characteristics of the included studies are shown in Table 1.

### 3.1. Risk of Bias of Included RCTs

In general, the included studies were judged to be at unclear risk of bias due to insufficient information regarding sequence generation, blinding of outcome assessment, and selective reporting. The majority of the studies were judged to be at high risk for blinding of participants and personnel. The majority of the studies were at high risk of bias or there was insufficient information to assess for incomplete outcome data and other bias. In most studies the role of the food industry was not clarified. The summary of the risk of bias across the included studies is shown in Figure 2 and Figure 3.

### 3.2. Descriptive Synthesis of Clinical Studies

Ten studies were from the WHO Western Pacific region [32,35,36,38,39,40,41,42,45,46] and eight studies were from Europe [33,34,37,43,44,47,48,49]. Most studies (15) were conducted at a city level. Eight studies included only women [35,37,39,40,43,46,47,48] including one in postpartum women [46] and five in postmenopausal women [35,39,40,43,48] six studies were on children [32,33,34,36], one only in boys [32] and two studies in boys and girls [42,45]. One study included only in adult men [41] and three studies included adult men and women [38,44,49].

The most common intervention food vehicles were milk, evaluated in eleven studies [35,36,38,39,41,42,43,45,46,47,48] and flour or derived products in four studies [32,33,44,49]. Only two studies used calcium salts as fortificants [44,49] (Table 1). Thirteen studies used milk mineral powders as fortificants, however the specific chemical composition of these powders were not specified in three studies [34,36,37] whereas seven reported that the milk powders contained phosphorous, potassium, magnesium and sodium [32,33,35,38,42,45,46], two studies fortified the food with calcium plus phosphorus [43,47] and one with calcium plus magnesium [39]. Two studies fortified the food with calcium and Vitamin D but only contributed for the calcium intake meta-analysis [41,48]. Finally one study did not specified how the soy drink was fortified with calcium [40].

Fortification level was less than 500 mg per day in five studies [36,40,47,48,49] between 500 and 1000 mg per day for five studies [32,34,35,41,44], and over 1000 mg per day in four studies [38,39,42,43] (Table 1). Three studies had two levels of intervention less than 500 mg per day and between 500 and 1000 mg per day [37,45,46], whereas one study had two levels of intervention less than 1000 mg per day and over 1000 mg a day [33].

Sixteen studies were trials of which twelve [32,33,34,36,37,39,40,42,43,45,46,47] were randomised parallel controlled trials, three crossover [35,38,44] and one 2 × 2 factorial [41]. Of the 16 RCTs, nine compared the intervention with the same unfortified food or the same food with lower calcium content [32,33,37,38,42,43,44,45,46], four compared the intervention with usual diet [35,36,40,41], one compared the intervention with a supplement [34] and one had high calcium milk as the intervention and used unfortified apple juice as the comparator [39]. Although Barnuevo is an RCT we used the information of the intervention arm as a before and after study [47]. We also included two before and after studies [48,49].

Considering an adequate intake above the IOM’s estimated average requirement (EAR) for calcium, two out of three studies including children aged 4 to 10 years, had adequate intake (EAR is 800 mg a day) [32,42]. None of the two studies including children aged 11 to 18 years, had adequate intake (EAR is 1100 mg a day) [34,45]. Two out of three studies including adults aged 19 to 50 years, had adequate intake (EAR is 800 mg a day) [37,44,46]. Four out of seven studies in adults aged 50 years or more had, adequate intake (EAR is 1000 mg a day for women and 800 mg a day for men) [35,38,39,41,43,48,49].

Four studies had a duration of less than 2 months [37,38,39,48], one study a duration between 2 and less than 6 months [44], three studies 6 to less than 12 months [32,33,43], five studies between 12 to less than 24 months [40,41,42,46,47] and five studies over 24 months [34,35,36,45,49].

The main outcome reported in the studies was bone health for fourteen studies [32,33,34,35,36,37,39,40,41,42,43,45,46,47], whereas two had dietary calcium intake [48,49], one cholesterol levels [44] and one blood pressure [38].

We found that ten studies with a duration from 1 to 72 months reported calcium intake [32,36,37,39,40,41,44,45,48,49], nine studies with a duration from 8.5 to 24 months reported anthropometric outcomes [32,33,34,35,36,42,43,45,46] four studies with a duration from 1 to 24 months reported outcomes related calcium metabolism [36,37,39,43] and finally ten studies with a duration from 8.5 to 24 months reported outcomes related to bone health [32,33,34,35,36,40,42,45,46,47].

### 3.3. Synthesis of Economic Studies

We included two economic studies, one from the United States and the other from Germany. The study of Keller et al., from the United States aimed to assess the cost of calcium from a wide variety of sources in diverse cities of United States, while controlling for seasonal variations [50]. The study of Sandmann et al., from Germany aimed to assess the cost of calcium from a wide variety of sources in diverse cities of Germany, while controlling for seasonal variations [51].

### 3.4. Effect of Interventions

We present the meta-analysis effect sizes, separately for RCT and Non- RCT. The Summary of findings is presented in Table 2.

### 3.5. Calcium Intake

Ten RCTs, lasting from 1 to 72 months, provided data for the calcium intake outcome [32,36,37,39,41,44,45,46,48,49]. Meta-analysis of three RCTs in children showed an increase of 306.17 mg of calcium intake per day (95% CI 198.9 to 413.4, three studies, I^2^ = 90%) in the calcium fortification group compared to the control group [32,36,45]. Three RCTs in adults showed an increase of 471.5 mg of calcium intake per day (95% CI 266.5 to 676.4, I^2^ = 89%) [37,41,46]. One RCT reported calcium intake in postmenopausal women with an estimated effect of 1210.0 mg of calcium intake increase per day (95% CI 1162.8 to 1257.2) [39].

Of the three non–RCT reporting calcium intake, two were in adults and showed an increase in calcium intake of 639.6 mg per day, (95% CI 67.0 to 1212.1, I^2^ = 89%) [44,49], and one study in postmenopausal women showed an increase of calcium intake of 103.1 mg per day, (95% CI 96.1 to 110.1) [48] (Appendix A).

The effect by calcium fortification level reported in RCTs showed a strong dose-response trend. A low calcium fortification level of 244 mg/day, increased calcium intake in 74.90 mg/day (95% CI –147.6 to 297.4), a medium fortification of 459–600 mg/day increased calcium intake in 258.1 mg/day (95% CI 218.7 to 297.5) and a high fortification of 676–900 mg/day increased calcium intake in 477.35 (95% CI 434.5 to 520.2): Test for subgroup differences: *p* < 0.00001, I^2^ = 96.7% (Appendix A).

### 3.6. Anthropometric Outcomes

The fortification intervention resulted on a body weight increase of 0.22 kg, (95% CI −0.95 to 1.38, I^2^ = 0%) in children, five RCTs with a duration from 8.5 to 24 months [32,33,34,42,45], whereas in postpartum women the increase was 1.85 Kg, (95% CI −0.94 to 4.64) one RCT with a duration of 12 months [46] and in postmenopausal women was −0.03 Kg, (95% CI −4.11 to 4.05) one RCT with a duration of 6 months [43] (Appendix A).

The estimated effect of the fortification intervention on height change in children was 0.83 cm, (95% CI 0.00 to 1.65 I^2^ = 0%), six RCT with a duration from 8.5 to 24 months [32,33,34,36,42,45] (Appendix A).

### 3.7. Outcomes Related to Calcium Metabolism

The estimated effect of the fortification intervention on plasma parathyroid hormone in children was −1.51 pmol/L, (95% CI −2.37 to −0.65) one RCT with a duration of 24 months [36]. In premenopausal women, the effect was 0.10 pmol/L, (95% CI −0.67 to 0.88) one RCT with a duration of 4 weeks [37] and in postmenopausal women was −0.28 pmol/L, (95% CI −0.83 to 0.27) one RCT with a duration of 4 weeks [39] (Appendix A).

The estimated effect of the fortification intervention on plasma 25 hydroxy cholecalciferol (25 (OH) D3 in children was −0.60 ng/mL, (95% CI −1.28 to 0.08) one RCT with a duration of 24 months [36], whereas in postmenopausal women was 5.03 ng/mL, (95% CI −3.08 to 13.14) one RCT one study with a duration of 6 months [43] (Appendix A).

The estimated effect of the fortification intervention on serum calcium in children was −1.60 mg/dL, (95% CI −1.79 to −1.41) one RCT with a duration of 24 months [36], whereas in post-menopausal women was −0.01 mg/dL, (95% CI −0.11 to 0.08, I^2^ = 41%) two RCTs with a duration of 4 weeks and 6 months [39,43] (Appendix A).

The estimated effect of the fortification intervention on urine calcium/creatinine ratio in children was −0.05, (95% CI −0.07 to −0.03) one RCT with a duration of 24 months [36], whereas in post-menopausal women was −0.01, (95% CI −0.07 to 0.05) one RCT one study with a duration of 6 months [43] (Appendix A).

### 3.8. Outcomes Related to Bone Structure

The estimated effect of the fortification intervention on Bone Mineral Density (BMD) Femoral neck in children was 0.02 g/cm^2^, (95% CI 0.01 to 0.04 I^2^ = 0%), four RCTs studies with five subgroups and a duration from 11 to 24 months [33,34,42,45], −0.01 g/cm^2^, (95% CI −0.04 to 0.03) in postpartum women, one RCT with a duration of 12 months [46] (Figure 4).

The estimated effect of the fortification intervention on BMD Femoral neck inpre-monopausal women was −0.01 g/cm^2^, (95% CI −0.05 to 0.03) in one Non-RCT with a duration of 18 months [47] (Figure 5).

The estimated effect of the fortification intervention on BMD Lumbar spine in children was 0.01 g/cm^2^, (95% CI −0.00 to 0.03, I^2^ = 0%) four RCTs with five subgroups and a duration from 11 to 24 months [33,34,42,45], −0.02 g/cm^2^, (95% CI −0.06 to 0.02) in postpartum women, one RCT with a duration of 12 months [46] (Appendix A).

The estimated effect of the fortification intervention on BMD Lumbar spine in premenopausal women was 0.00 g/cm^2^, (95% CI −0.05 to 0.05) in one Non–RCT study with a duration of 18 months [47] (Appendix A).

The estimated effect of the fortification intervention on BMD hip in children was 0.03 g/cm^2^, (95% CI 0.00 to 0.06, I^2^ = 0%) in two RCTs with a duration of 18 to 24 months [42,45], −0.00 g/cm^2^, (95% CI −0.04 to 0.03) in postpartum women, one RCT with a duration of 12 months [46] (Appendix A).

The estimated effect of the fortification intervention on BMD hip in pre-menopausal women was −0.01 g/cm^2^, (95% CI −0.05 to 0.03) in one Non–RCT study with a duration of 18 months [47] (Appendix A).

The estimated effect of the fortification intervention on BMD total body in children was 0.01 g/cm^2^, (95% CI 0.00 to 0.02, I^2^ = 0%) three RCTs with a duration from 18 to 24 months [36,42,45], whereas in postpartum women was 0.01 g/cm^2^, (95% CI −0.02 to 0.03) one RCT with a duration of 12 months [46] (Appendix A).

The estimated effect of calcium fortification intervention on BMC total body in children was 11.59 g, (95% CI −36.55 to 59.74, I^2^ = 0%) four RCTs with a duration of 1 to 18 months [32,34,42,45] (Appendix A).

The estimated effect of calcium fortification intervention on BMC lumbar spine in children was 0.69 g, (95% CI −0.53 to 1.90, I^2^ = 0%) three RCTs with four subgroups with a duration of 1 to 24 months [32,42,45] (Appendix A).

The estimated effect of calcium fortification intervention on BMD Trochanteric region in children was 0.03, (95% CI −0.00 to 0.07) one RCT with a duration of 18 months [42]. The estimated effect of the calcium fortification intervention on BMD Trochanteric in adults was −0.00, (95% CI −0.03 to 0.03) one RCT with a duration of 12 months [46] (Appendix A).

The estimated effect of the calcium fortification intervention on BMD Trochanteric region in adults was −0.00, (95% CI −0.04 to 0.04) one Non–RCT study with a duration of 18 months [47] (Appendix A).

### 3.9. Other Outcomes

The following outcomes were reported in only one RCT showing no effect of calcium fortification intervention, total cholesterol −0.20, (95% CI −0.61 to 0.21); LDL cholesterol −0.20, (95% CI −0.51 to 0.11), HDL cholesterol (MD 0.00, 95% CI −0.20 to 0.20), LDL/HDL Ratio −0.10, (95% CI −0.42 to 0.22), TAG 0.10, (95% CI −0.17 to 0.37; participants = 0; studies = 1) Office sitting 0.00, (95% CI −6.31 to 6.31; participants = 0;) [44] (see Appendix A). 

Appendix A present the meta-analyses for the effect of calcium fortification on calcium intake (mg/day), BMD femoral neck and femoral shaft (g/cm^2^), body mass index (kg/m^2^), bmc hip, Femoral neck and Femoral shaft (g), calcium intake by calcium fortification level (mg/day), parathyroid hormone by calcium fortification level (pmol/L), serum 1,25-dihydroxycholecalciferol by calcium fortification level (nmol/L), weight by calcium fortification level (Kg), BMD femoral neck, lumbar spine, hip and total body by calcium fortification level (g/cm^2^). None of the included studies reported preeclampsia, cardiovascular outcomes, hypertension or lithiasis.

Three RCTs were not included in the meta-analysis. Green et al. was the only trial reporting blood pressure. The four-week cross over RCT assessed the effect of a commercially available high calcium skim milk powder providing 1075 mg of calcium in 50 g of powder compared to normal skim milk powder with 720 mg of calcium in 50 g of powder in the blood pressure of 38 men and women aged over 40 years [38]. The two powder milks differ slightly in the magnesium and sodium composition. High calcium skim milk powder contained 74 mg of magnesium and 208 mg of sodium per 50 g of powder compared to 64 mg of magnesium and 197 mg of sodium per 50 g in the normal skim milk. Office sitting systolic blood pressure did not change after either SMP (121 ± 14 mmHg at start versus 122 ± 15 mmHg at the end or high-calcium SMP 125 ± 19 mmHg at start versus 122 ± 13 mmHg at the end. Gui et al. assessed in a 18 months open label RCT the effect of a 250 mL of milk providing 100 mg of calcium in 100 mL and 250 mL of fortified soymilk with 100 mg of calcium in 100 mL compared to normal diet in bone mineral of 141 postmenopausal women aged 45 to 55 years. We did not include this study in the meta-analysis as it compared two foods with similar calcium content. Besides the milk and soymilk differ in fat composition, soymilk contained 1 g of fat compared to 5 g of fat in the milk [40]. Daily consumption of cow’s milk containing significantly increased the BMD in the femoral neck and in the total hip. On the other hand, daily consumption of soy milk had no effect. The BMD in the hip (2.52%) and the femoral neck (2.82%) of the women consuming milk was significantly higher (hip, *p* = 0.01; femoral neck, *p* < 0.0000001). The women in the control group experienced a reduction in BMD at all sites; the reduction in BMD was only significant at the hip during 12 months (*p* = 0.008) and at the femoral neck during 18 months (*p* = 0.005). Finally, the study of Cleghorn [35] was excluded from the meta-analysis as we found inconsistencies in the reporting of the results. The study evaluated in a 2-year open cross over study the effect of a calcium-enriched milk on bone loss in women who are within five years of the menopause and has a basal calcium intake of 1250 mg or higher. The calcium-enriched milk had an extra 685 mg of calcium and 514 mg phosphorous. The main outcomes were forearm and lumbar spine BMD however the results were inconsistent. The effect on bone loss from the spine was 1.76 percentage points less when the women were taking the milk supplement than when they were on their usual diet (95% CI, 0.54–2.98%; *p* = 0.006).

### 3.10. Sensitivity and Subgroup Analysis

We explored the potential effect of phosphorus as a cointervention in all the outcomes and we did not find any difference between those with and without phosphorous as cointervention. We did not find important differences in the effect of fortification in the subgroup of studies before and after one year of intervention.

### 3.11. Economic Studies

Two studies were included for the economic synthesis. Keller et al. found that the cost of estimated absorbable calcium in the United States ranged widely from one product to another. The study found that the least expensive sources of calcium were cereal, skim milk and calcium-fortified orange juice from frozen concentrate, whereas other dairy products, such as mozzarella and low-fat yogurt, were considerable more expensive sources [50].

Sandmann et al. showed that the total costs of an hypothetical vitamin D and calcium voluntary food fortification programme amounted to €41 million per year in Germany (currency euro 2013 year 2014): €33.1 million for cholecalciferol and calcium, €3.3 million for marketing and education activities, €2.9 million for food control and monitoring, and €2.1 million for other programme-specific recurrent production costs. On the other hand, €356 million in the cost of fractures were saved per year [51]. The largest cost savings (43% of the total cost savings) came from prevented hip fractures, with savings of €152·5 million, while the other cost savings came from averted fractures of the humerus (€61.8 million, 17%), clinical vertebral (€46·4 million, 13%), pelvis (€39·2 million, 11%), wrist (€34.4 million, 10%) and other femur (€21·9 million, 6%).

## 4. Discussion

We found that fortification increased calcium intake in all studied age groups (moderate certainty-evidence) for children and low certainty-evidence for adults, indicating that fortified foods could be an effective way to help populations attain dietary calcium recommendations [6]. Food fortification with calcium increased height in children (moderate certainty-evidence), however no changes were observed in weight or BMI. We found marginal effects with limited clinical relevance in children’s BMDs total body, lumbar spine, femoral neck moderate certainty-evidence and hip, five studies with a duration from 11 to 24 months [33,34,36,42,45]. No clinically or statistically significant effects were found in RCTs on weight (moderate certainty-evidence), BMI, bone health, or metabolic parameters of adults and no effects were observed in non-randomised studies.

Calcium balance is hence actively controlled by a large number of factors. The external balance of calcium (the difference between intake and output) is, in effect, determined by the exchange between the skeleton, the intestine and the kidney. These fluxes are controlled by the action of calciotrophic hormones: parathyroid hormone; 1,25-dihydroxycholecalciferol; calcitonin. It is also influenced by other factors such as sex hormones, growth hormones, corticosteroids and a variety of locally-acting hormones. Vitamin K supplementation at doses obtainable in the diet from a 50 g portion of green leafy vegetables, in combination with calcium and vitamin D has been suggested to play a role in the optimization of bone health. We found one study reporting a decrease in PTH blood levels, calcaemia and urine calcium creatinine ratio in girls with low calcium intake and high prevalence of Vitamin D deficiency receiving a fortified milk compared to usual diet [36]. It is documented that an increased calcium intake inhibits PTH release from the parathyroid glands, and this has been linked to lowering blood pressure and reducing weight [13,52].On the other hand, the explanation for the decrease in urine calcium creatinine ratio and calcaemia in the fortified group is less clear Du et al. postulate that the fortification increased the lean tissue and thus urine creatinine excretion that possibly masked the increase in urine calcium in those girls receiving the intervention [36]. Similarly, the decrease in calcaemia found in the fortified group of Du et al. study could be explained by an increase in calcium incorporation in the bone.

Importantly, none of the included RCTs reported adverse effects of calcium fortification alone or in combination with other minerals. However, the evidence we found in this review should be taken with caution as most RCTs were assessed as low quality, mainly due to lack of blinding and attrition bias. Except from the outcomes ‘calcium intake’ in adults and postmenopausal women, in which the certainty of evidence was low, for the remaining clinical outcomes considered, it was deemed moderate.

Most food vehicles used for fortification identified in this review were dairy products, mainly milk. Other food vehicles were flour and bakery products. Most fortificants used to add calcium were milk-derived powders or salts, which besides calcium, added other minerals such as phosphorus and magnesium to the fortified foods. Phosphorus is known to participate in the calcium metabolisms and magnesium has been reported to also have effects on blood pressure and bone health [45]. Therefore, this evidence may be less relevant or difficult to apply in those countries where dairy products are not regularly consumed which are the vast majority of LIC [53].

Most of the identified evidence came from upper-middle income countries that mostly had adequate calcium intake. This could be one of the reasons of the lack of important effects. Other reasons for absence of effect could be study duration, in children the duration was at least 12 months in four out of six studies, in adults there were four out of nine studies that lasted at least 12 months. Another reason could be the low fortification level, as some studies had a comparison group that also added calcium, leaving a net difference between the intervention and the comparison groups of less than 600 mg of calcium a day in eight out of 15 RCTs and only one more than 1000 mg a day.

Evidence from market derived fortification in the US shows that the least expensive foods to fortified are cereals, skimmed milk and orange juice [50]. From a national fortification program in Germany we only found that the largest savings came from hip fracture prevention in older women with a total saving cost of €315 million and prevention of 36 705 fractures in the target population [51].

### 4.1. Strengths

We did an exhaustive search using a broad search strategy in most important databases following Cochrane methods and PRISMA statement for reporting. We also carefully explored grey literature and the reference lists in systematic reviews to detect more studies. We performed the risk of bias assessment and the Grade analysis and sensitivity analysis for subgroup.

### 4.2. Limitations

The main limitation of the review is the difficulty to isolate the effect of calcium alone. The evidence for food fortification with calcium alone is scarce so the effects we show are for calcium in combination with other minerals mainly those from milk such as phosphorus and magnesium that can also influence the health outcomes we studied. Although we did sensitivity analysis for those that had calcium alone as the intervention or combined interventions and found no difference. Besides, although the studies report the calcium fortification level provided per daily portion of food, the actual food intake was only reported by six of the 15 included RCTs. Another limitation is that we could not draw reliable conclusions from various subgroup analyses due to the restricted number of studies in each subgroup. In addition, most food vehicles evaluated were dairy products which are not highly consumed in many LMICs where most populations with low calcium intake are found.

The documented evidence of the effects of calcium food fortification seems scarse, despite the existance of many calcium fortified foods in the market. This is in agreement with other recent reviews on the topic [54]. Food fortification strategies have been implemented for more than 80 years and have contributed to an improvement in health [55]. Nowadays, more than 130 countries have mandatory food fortification with micronutrients such as iodine, iron, iodine, folate, and vitamin A, however only one country, the UK, has mandatory flour fortification with calcium [56,57]. Even this, the existance of mandatory fortification of foods with other minerals in several countries provides a technological and legal framework that could facilitate the addition of calcium [58].

Except for the economic evaluation and the before and after study from Osler et al. we found no evidence from large scale fortification programs which are important to measure the effect in real conditions [51]. We did not find any study for some age groups such as men alone and adolescents that have evaluated clinical outcomes, therefore further studies are needed to assess the calcium fortification effectiveness and safety for these populations. Calcium fortification studies should also confirm the be benefits in older adults as the calcium supplementation effects is controversial [59,60].

## 5. Conclusions

This review suggests that calcium fortification increased calcium intake in a dose-response fashion, suggesting a good acceptability and a feasible method to reach dietary recommendations. Even if the effects on height and bone health in children were found small, they deserve further considerations, due to its potential public health impact in the prevention of stunting and bone diseases. We did not find any statistically significant effect on weight, BMI, bone health, or metabolic parameters of adults. Our review highlights a substantial evidence gap on the effects of calcium fortified foods on blood pressure and bone fracture. There is a lack of studies on adolescents, men and pregnant women exploring the effect on blood pressure, bone health and weight. In addition, studies are needed to explore the effect of higher fortification levels, more appropriate comparison groups in the context of better powered trials and longer intervention periods as there is evidence that there could be a not long-lasting effect [61,62]. High-quality RCTs, representative quasi-experimental studies, including large-scale fortification programs, and studies using higher fortification doses are required to confirm if the small but significant effects we found can reach clinically relevant effects. There is scarce information on the sustainability of these programmes in the long-term.

## Figures and Tables

**Figure 1 nutrients-13-00316-f001:**
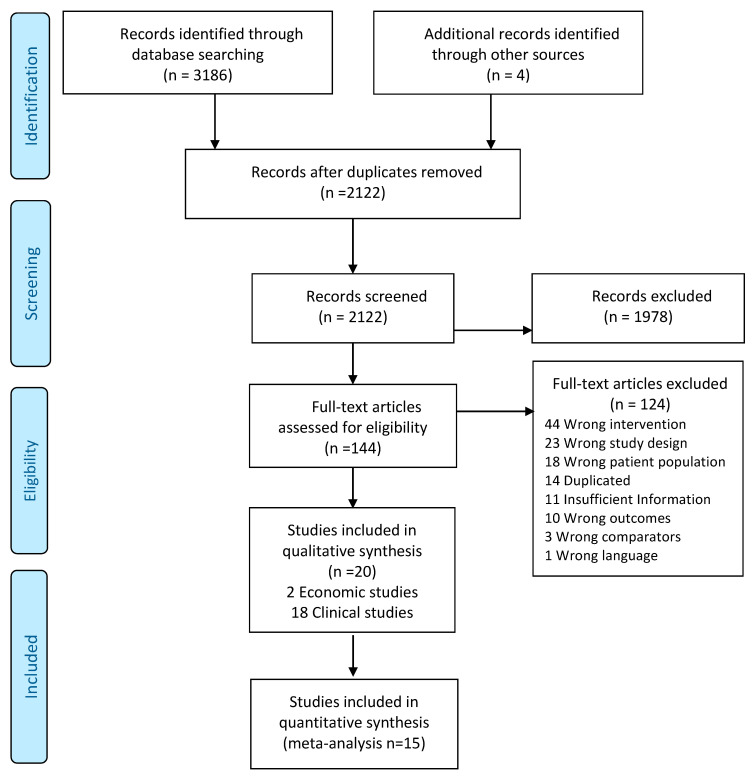
Study flow diagram.

**Figure 2 nutrients-13-00316-f002:**
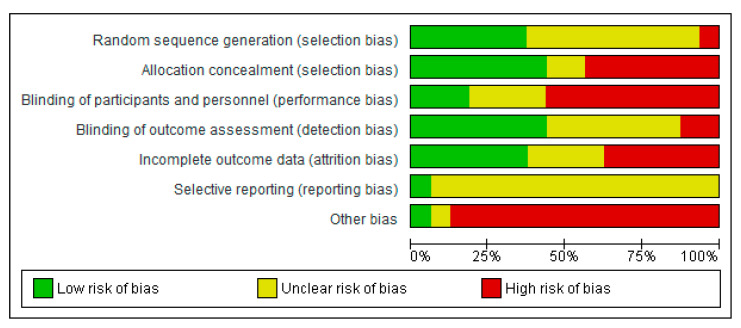
Review authors’ judgements about each risk of bias item presented as percentages across all included Randomised Controlled Trials.

**Figure 3 nutrients-13-00316-f003:**
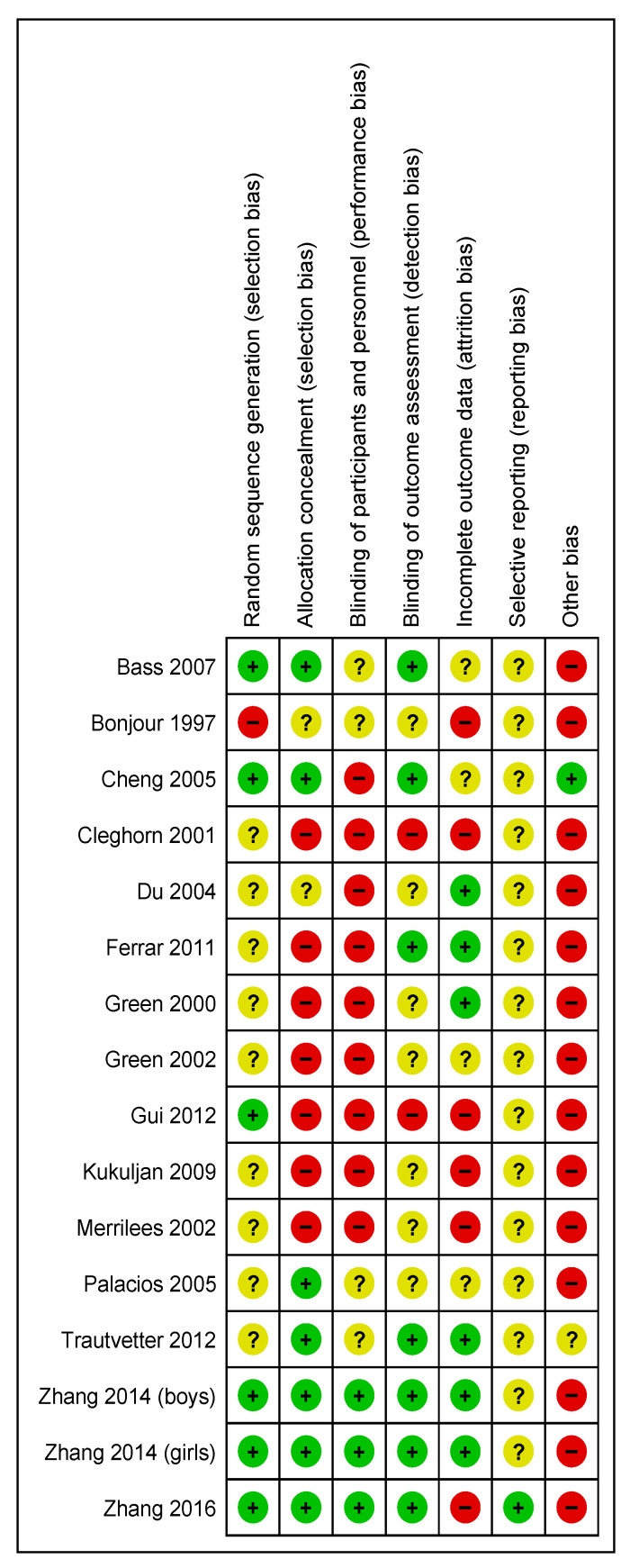
RCT risk of bias summary for included Randomised Controlled Trial [32,33,34,35,36,37,38,39,40,41,42,43,44,45,46].

**Figure 4 nutrients-13-00316-f004:**
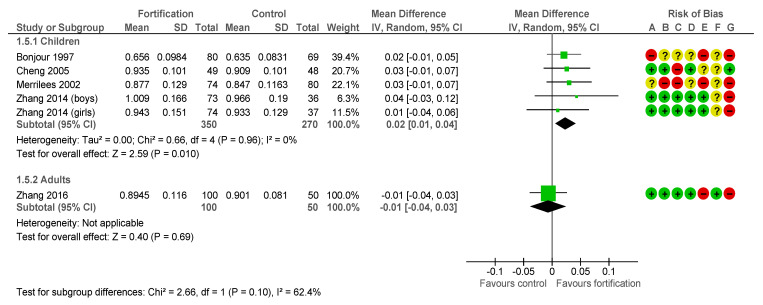
Impact of calcium fortification intervention on BMD femoral Neck (g/cm^2^) in RCTs [33,34,42,45].

**Figure 5 nutrients-13-00316-f005:**
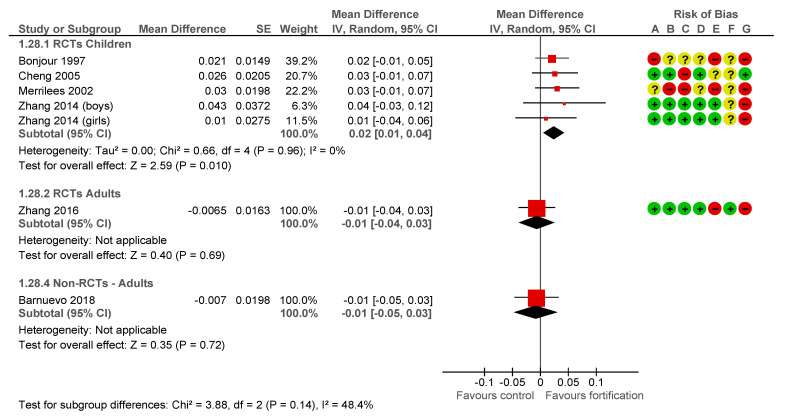
Impact of calcium fortification intervention on BMD femoral Neck (g/cm^2^). Generic inverse variance [33,34,42,45,46,47].

**Table 1 nutrients-13-00316-t001:** Characteristics of the included studies.

Author	Region	Country	Population Characteristics	*N*	Study Design	Fortified Food	Baseline Calcium Intake mg/day	Intervention and Control Group Used for this Analysis	Food Fortification Level < 0.5; 0.5–1; >1 g/assigned Daily Portion	Calcium Intake Difference between Intervention and Control Group (mg/day)	Duration of Fortification (Months)	Main Outcome	Outcomes Assessed Included in This Review
Bass 2007 [32]	Western Pacific Region	Australia	Healthy boys between 7–11 years old	88	RCT	Cakes/cookies	I: 931 P:934	Intervention: one out of 10 varieties of muffins and cookies a day. Muffins and cookies were fortified with 4 g of milk minerals powder, which provided participants with an additional 800 mg of elemental calcium per day. Placebo: one out of 10 varieties of muffins and cookies without added calcium.	0.5–1	800	8.5	Bone health	Calcium Intake, Weight, Height, BMC (total body, lumbar spine, leg (Femur + Tibia-Fibula), arm (Humerus + Ulna-Radius)
Bonjour 1997 [33]	Europe	Switzerland	Healthy prepubertal Caucasian girls. Mean age 7.93 years old	149	RCT	Cakes	NA	Intervention: Two food products every day in place of similar foods taken for breakfast or snacks fortified with calcium from milk extract. The calcium contents (mg/serving) of calcium-enriched food products was as follows: chocolate cakes, 516; caramel cakes, 512; biscuits, 548; fruit juices, 383; powdered drinking chocolate, 530; chocolate bars, 429; yogurts, 478. Calcium from milk extract was used to fortify food products. Placebo: Two food products every day in place of similar foods taken for breakfast or snacks without added calcium. The calcium contents (mg/serving) of placebo food products was as follows: chocolate cakes, 33; caramel cakes, 41; biscuits, 8; fruit juices, 11; powdered drinking chocolate, 4; chocolate bars, 80; yogurts, 174.	0.5–1/>1	-Chocolate cakes, 483 -Caramel cakes, 471 -Biscuits, 540 -Fruit juices, 372 -Powdered drinking chocolate, 526 -Chocolate bars, 349 -Yogurts, 304	11	Bone health	Weight, Height, BMI, BMD (lumbar spine, femoral neck)
Cheng 2005 [34]	Europe	Finland	Girls aged 10–12 years old	195	RCT	Cheese	I: 706 P: 664	Intervention: dairy products such as natural low-fat cheese (110 g Edam with 17% fat and 100 g Turunmaa with 15% fat) which provided a quantity equivalent to 1000 mg of elemental calcium a day. Placebo: calcium plus vitamin D supplements placebo.	0.5–1	1000	24	Bone health	Weight, Height, aBMD (lumbar spine, femoral neck, total femur), BMC total body.
Cleghorn 2001 [35]	Western Pacific Region	Australia	Women who were less than five years postmenopausal.	142	RCT-c	Milk	I: 967 C: 918	Intervention: one litre of calcium-fortified milk thrice weekly (428 mL/d). Milk was fortified by adding to reduced-fat milk the retentate from ultrafiltration of low-fat milk. It contains 1600 mg of elemental calcium and 1200 mg phosphorus per litre providing an average of 685 mg of elemental calcium a day. Control: Usual diet	0.5–1	686	24	Bone health	Weight, BMD (Forearm, Lumbar spine L2–L4)
Du 2004 [36]	Western Pacific Region	China	Healthy girls aged 10 years old	757	RCT	Milk	I: 418.2 C: 455.3	Intervention: 330 mL ultra-heat-treated (UHT) milk 5 days a week, which had been fortified to contain 560 mg of elemental calcium. It provided 245 mg of elemental calcium a day. Control: Usual diet	<0.5	245	24	Bone health	Calcium Intake, Height, PTH, Plasma 25(OH)D, Plasma Ca, BMD Total body, BMI, Urine Ca/creatinine.
Ferrar 2011 [37]	Europe	UK	Young women ages 20 to 39 years old	76	RCT	Ice cream	I 1: 735 I 2: 663 I 3: 754 P: 714	One per day Ice cream low in fat—Calcium, magnesium, phosphorus and zinc. Milk minerals (Arla Foods Ingredients, Redhill, Surrey, UK) Intervention Group 1: 60 g ice cream containing 244 mg of elemental Calcium fortified with milk minerals. Intervention Group 2: 60 g ice cream containing 459 mg of elemental Calcium fortified with milk minerals. Intervention Group 3: 60 g ice cream containing 676 mg of elemental Calcium fortified with milk minerals. Placebo: consumed 60 g of ice cream containing 96 mg calcium per unit.	<0.5/0.5–1	I 1: 148 I 2: 363 I 3: 580	1	Bone health	Calcium Intake, PTH, Serum 1,25D.
Green 2000 [38]	Western Pacific Region	New Zealand	Healthy volunteers aged over 40 years, 19 men and 19 women.	38	RCT-c	Milk	1120	A 4-week washout period separated consecutive milk interventions. Each volunteer consumed each of the milks in randomized order. Intervention: 50 g of high Calcium skim milk powder diluted in tap water to provide 480 mL of milk per day containing 1075 mg of elemental Calcium. Control: 50 g of skim milk powder diluted in tap water to provide 480 mL of milk per day containing 720 mg of Calcium.	>1	355	1	Blood Pressure	Office sitting SBP (mmHg), office sitting DBP (mmHg), office standing SBP (mmHg), office standing DBP (mmHg).
Green 2002 [39]	Western Pacific Region	New Zealand	Healthy postmenopausal women (at least 5 years postmenopausal)	50	RCT	Milk	I: 850 C: 900	Intervention: 400 mL of high-calcium skim milk powder containing 1200 mg of calcium supplemented with 172 mg magnesium per 50 g milk powder a day. Control: 400 mL of apple drink containing no more than 25% apple juice a day.	>1	1200	1	Bone Health	Calcium Intake, Serum PTH (pmol/L), Serum Calcium.
Gui 2012 [40]	Western Pacific Region	China	Postmenopausal women without osteoporosis, aged 45–65, and postmenopausal for more than 2 years.	141	RCT	Soymilk	NA	Intervention: 250 mL calcium-fortified soymilk daily. Calcium-fortified soymilk contained 6.5 g soy protein, 2.5 g fat, 2.5 g lactose, 250 mg calcium, and 3.75 to 4.5 mg soy isoflavones. Control: Usual diet. They abstain from any other dietary supplementation, including other milk, other soymilk, vitamin D, vitamin K, complex vitamins, and calcium tablets.	<0.5	250	18	Bone Health	BMD (spine, femoral neck, hip).
Kukuljan 2009 [41]	Western Pacific Region	Australia	Healthy community- dwelling Caucasian men aged 50–79 years	180	RCT- 2 by 2factorial design	Milk	I: 1039 C: 996	Intervention: 400 mL milk per day of reduced-fat (1%) ultrahigh temperature (UHT) milk. Milk was fortified with milk salts containing 1000 mg calcium and 800 IU vitamin D3 and 500 mg phosphorous/day Control: Usual Diet.	0.5–1	1000	18	Bone Health	Calcium intake
Gibbons 2004 [42]	Western Pacific Region	New Zealand	Children, aged 8–10 years	154	RCT	Milk	I: 934 P: 985	Intervention: 8O g of chocolate milk drink per day. The high calcium milk provided 1200 mg of elemental calcium and 776 mg phosphorus per day. Placebo: 8O g of chocolate milk drink per day. The milk provided 400 mg of elemental calcium and 320 mg of phosphorus per day.	>1	800	18	Bone Health	Weight, Height, BMD (Lumbar spine, femoral Neck, total hip, trochanter, total body) and BMC (total body, trochanter, hip, femoral neck and spine)
Palacios 2005 [43]	Europe	Spain	Healthy white women, postmenopausal for 10 years or more, between 49 and 71 years old and with a dietary calcium intake lower than 750 mg/day	79	RCT	Milk	I: 508 P: 502	Intervention: 750 mL of skimmed milk enriched with calcium 1200 mg, phosphorus 945 mg, lactose, and vitamin D3 5.7 mg per day. Placebo: 750 mL skimmed milk enriched with vitamin D3 (5.7 mg/750 mL) of identical appearance, taste, and composition to that of intervention group except for the amount of calcium (900 mg/750 mL) and this milk was not fortified with phosphorus or lactose.	>1	300	6	Bone Health	Weight, 250H vitamin D3, Calcemia (mg/dL), calcium urine (mg/dL), calcium/creatinine (mg/mg)
Trautvetter 2012 [44]	Europe	Germany	Men and women omnivorous, moderately hypercholesterolemic subjects; aged 25.5 y and had a BMI of 22.3 kg/m^2^.	32	RCT-c	Bread	873	All subjects consumed 100 mL of the probiotic drink daily. Intervention: 135 g of bread a day fortified with 1000 mg of elemental calcium as pentacalcium hydroxy- triphosphate Control: 135 g of bread a day without added calcium.	0.5–1	1000	3	Cholesterol	Calcium Intake, Total cholesterol [mmol/L], LDL-cholesterol [mmol/L], HDL-cholesterol [mmol/L], LDL/HDL ratio, triacylglycerol [mmol/L]
Zhang 2014 [45]	Western Pacific Region	China	Healthy adolescents aged 12–14 years (111 girls and 109 boys)	220	RCT	Milk	Girls: I 1: 651 I 2: 707 C: 701 Boys: I 1: 758 I 2: 704 C: 680	The subjects were assigned to receive 40 g of milk powder daily. Each daily dose was administered in two packages (approximately 20 g/package). Intervention 1: 40 g of milk powder containing 900 mg of calcium and 200 IU of vitamin D. For this fortification daily dose of 2g of isolated milk salt (containing 29.2% calcium and 15% phosphorus) was added to 40 g of milk. Intervention 2: 40 g of milk powder containing 600 mg of calcium and 200 IU of vitamin D. For this fortification daily dose of 1g of isolated milk salt (containing 29.2% calcium and 15% phosphorus) was added to 40 g of milk. Control: 40 g of milk powder containing 300 mg of calcium and 200 IU of vitamin D.	<0.5/0.5–1	I 1: 600 I 2: 300	24	Bone Health	Calcium Intake, Weight, Height, BMD and BMC (spine, femoral neck, left hip, total body, femoral shaft)
Zhang 2016 [46]	Western Pacific Region	China	Postpartum women aged 20–35 years. All were primipara who had delivered a normal single infant at full term and intended to be breast-feeding	150	RCT	Milk	I 1: 822 I 2: 811 C: 807	The subjects were assigned to receive 40 g of milk powder daily for 12 months. Each daily dose was administered in two packages (approximately 20 g/package). Intervention 1: 40 g of milk powder containing 900 mg of calcium and 5 μg of vitamin D. For this fortification daily dose of 2g of isolated milk salt (containing 29.2% calcium and 15% phosphorus) was added to 40 g of milk. Intervention 2: 40 g of milk powder containing 600 mg of calcium and 5 μg of vitamin D. For this fortification daily dose of 1g of isolated milk salt (containing 29.2% calcium and 15% phosphorus) was added to 40 g of milk. Control: 40 g of milk powder containing 300 mg of calcium and 5 μg of vitamin D.	<0.5/0.5–1	I 1: 600 I 2: 300	12	Bone Health	Calcium Intake, Weight, BMD (spine, femoral neck, left hip, trochanter, total body)
Barnuevo 2018 [47]	Europe	Spain	Healthy female young volunteers. Mean age 39.2 ± 4.6 years old	181	RCT analyzed as UBAS	Milk	NA	Intervention: 250 mL a day of partly skimmed milk with 240 mg calcium and 105 mg of phosphorus.	<0.5	NA	18	Bone health	BMD (lumbar spine, femoral neck, total hip, throcanteric region)
Gonzalez Sanchez 2012 [48]	Europe	Spain	Postmenopausal women, aged between 36 and 84 years, and who had low intake of calcium and vitamin D.	261	UBAS	Fermented Milk	747.9	Intervention: 125 g of Fermented milk (Densia^®^) per day, fortified with 400 mg of elemental calcium and 200 UI of Vit D.	<0.5	400	1	Dietary Intake	Calcium intake
Osler 1998 [49]	Europe	Denmark	Men and women, aged 35–65 years at first examination in 1987, 1988	329	UBAS	Flour	1215	Intervention: Flour fortified with calcium, 200 mg of elemental calcium per 100 g wheat flour and 400 mg of calcium per 100 g rye flour, since 1954, and until 1987 when the mandatory fortification was stopped.	<0.5	NA	72	Dietary Intake	Calcium intake

RCT: Randomized controlled trial; RCT-C: Cross over Randomized controlled trial; UBAS: Uncontrolled before-after study; NA: Not available; BMD: bone mineral density; BMC: bone mineral content; BMI: body mass index (w/h^2^); PTH: Parathyroid hormone; SBP: Systolic blood pressure; DBP: Diastolic blood pressure; LDL: low-density lipoprotein; HDL: High-density lipoprotein.Excluded studies. Out of the 124 excluded studies, 7 were initially included so we detailed the final exclusion reason. Two did not have the intervention under study, one had a non-eligible comparison group and four a non-eligible study design. The reasons for exclusion are provided in Appendix B
Table A2)

**Table 2 nutrients-13-00316-t002:** Summary of findings of calcium fortification (+minerals/other) by population group.

Outcomes	Mean Difference (MD) of Fortified Versus Control Group * (95% CI)	No. of Participants (Studies)	Certainty of the Evidence (GRADE)
Calcium Intake (mg/day)—Children	MD 306.17 higher 198.97 higher to 413.38 higher)	764 (4 RCTs)	⨁⨁⨁◯ MODERATE ^a^
Calcium Intake (mg/day)—Adults	MD 471.47 higher (266.51 higher to 676.42 higher)	315 (3 RCTs)	⨁⨁◯◯ LOW ^a^^,^^b^
Calcium Intake (mg/day)—Postmenopausal Women	MD 1210 higher (1162.8 higher to 1257.2 higher)	50 (1 RCT)	⨁⨁◯◯ LOW ^c^
Weight (kg)—Children	MD 0.22 higher (0.95 lower to 1.38 higher)	667 (6 RCTs)	⨁⨁⨁◯ MODERATE ^d^
Weight (kg)—Adults	MD 1.85 higher (0.94 lower to 4.64 higher)	150 (1 RCT)	⨁⨁⨁◯ MODERATE ^e^
Weight (kg)—Postmenopausal Women (Vit D cointervention in both groups)	MD 0.03 lower (4.11 lower to 4.05 higher)	79 (1 RCT)	⨁⨁⨁◯ MODERATE ^e^
Height (cm)—Children	MD 0.83 higher (0 to 1.65 higher)	1164 (7 RCTs)	⨁⨁⨁◯ MODERATE ^f^
BMD Femoral neck (g/cm^2^)—Children	MD 0.02 higher (0.01 higher to 0.04 higher)	620 (5 RCTs)	⨁⨁⨁◯ MODERATE ^f^
BMD Femoral neck (g/cm^2^)—Adults	MD 0.01 lower (0.04 lower to 0.03 higher)	150 (1 RCT)	⨁⨁⨁◯ MODERATE ^g^

Quality of Evidence Symbol: ⨁⨁⨁◯ Moderate, ⨁⨁◯◯ Low. * Studies that compared the fortified food with similar unfortified food, with a food with lower content of calcium, with usual diet, or with supplement placebo; ^a^. I2 > 70% but not important clinical heterogeneity; ^b^. 2/3 studies with high Risk of Bias (RoB) for allocation concealment and blinding of participant and personnel; ^c^. The only study for this subgroup has high RoB for allocation concealment and blinding of participant and personnel; ^d^. 3 out of six studies present moderate or high RoB for most domains; ^e^. Wide 95% confidence interval; ^f^. Most studies present moderate or high RoB for many domains; ^g^. This is a single study with high RoB in the domain incomplete outcome data. Effects in primary and secondary outcomes are presented in Appendix A.

## Data Availability

Data is contained within the article or Appendix A.

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
