# Peer review of "Effect of Calcium Fortified Foods on Health Outcomes: A Systematic Review and Meta-Analysis"

_nutrients, 2021, doi:10.3390/nu13020316_

Round 1

Reviewer 1 Report

In the Introduction authors assumed only positive calcium supplementation effects. There is no discussion about negative effects of the supplementation, especially among elderly people when very often calcium supplementation is combined with vitamin D supplementation and vitamin K deficiency occurs.

Citation 5 from 1992 does not add anything to the discussion

Line 61 - RRR abbreviation is explained in line 63-64 and should be before

59-60 – sentence „Calcium effects in other groups …” is in present tense. Newberry et al. (autors of those publication) point out that this data should be cite carefully. Especially that there are newer ones.

57-59 „The benefits of calcium supplementation …” - missing info how often and which regions

72-73 - sentence „… as there are major acceptability and feasibility concerns …” is unclear. What is „acceptability” and „feasibility”? This sentence should be rewritten or split.

74-80 – no citation. There is too much information in this paragraph - the authors write both about the effects of an overdose of calcium (this requires expansion) and the economic costs. And at the end of the paragraph there is information that this applies to low- and middle-income countries. Could the side effects not occur in highly developed countries?

85 – it should be clarified what is „remote populations”

115-117 „The only case in which we accepted … no evidence of effect modification” – it is no clear what effect modification is about, this whole sentence is unclear  

128 – in this place authors indicate that language was not limitation, but 1 article was excluded before language (Figure 1) – for what reason this article was excluded?

155 – Statistical analyses - Did the authors check the possibility of increasing the confidence interval (e.g. up to 97%) in order to narrow the obtained numerical ranges, because for many data there is a very wide range, e.g. lines 276-277 (95% CI 147.6 to 297.4)

173 - the authors write that 5 articles were subject to narrative synthesis, but does not write about it in the methodology. It is unclear what exactly was narrative synthesis, this term may not be clear and requires explanation

226-232 - the authors refer the consumption of Ca to the official recommendations, but do not cite any of them, e.g. Dietary Reference Intakes for Calcium and Vitamin D, Institute of Medicine US, especially as the recommendations for the elderly differ slightly between countries

422-431 – the relationship between decreased PTH level and higher calcium intake should be better described, because the fact is that increase of Ca serum concentration increases PTH secretion. The authors explain that there may have been an increased incorporation of Ca in the bone, so it should be discussed in the context of vitamin K, which directly participates in this process

432-434 - it seems that the bias conclusion should refer to all results obtained and discussed, not just "no adverse effects of fortification"

Because calcium absorption is related to vitamin D and in some studies included to this meta-analysis was also analyzed supplementation with this vitamin, the authors should additionally compare the results from studies where only calcium was used with those where there was calcium + vitamin D (and of course this should be discussed). If this is impossible, there should be information about it, but it should be discussed anyway

If extended the manuscript vitamin D information would exceed the required volume of the manuscript, then you can resign from such extensive data on the intake of Ca, - it is obvious  that the consumption of Ca was higher in the supplemented groups.

For many reference items DOI is missing, although it exist, e.g. item 6, 7, 8, 12, 14, 15

Author Response

Reviewer 1

Comments and Suggestions for Authors

In the Introduction authors assumed only positive calcium supplementation effects.

There is no discussion about negative effects of the supplementation, especially among elderly people when very often calcium supplementation is combined with vitamin D supplementation and vitamin K deficiency occurs.

Response: We have added a phrase regarding recommendations in elderly adults.

Introduction:

“On the other hand, as calcium supplements has been related to constipation, bloating and kidney stones, and some evidence suggests they may cause a small increase in the risk of myocardial infarction in elderly adults a recent review does not recommend the use of calcium supplements in healthy community-dwelling adults.(Reid and Bolland, 2019)”

Citation 5 from 1992 does not add anything to the discussion

Response: Thank you we have move the citation to the corresponding phrase.

Line 61 - RRR abbreviation is explained in line 63-64 and should be before    

Response: Thank you we have move the RRR explanation.

59-60 – sentence ”Calcium effects in other groups …” is in present tense. Newberry et al. (authors of those publication) point out that this data should be cite carefully. Especially that there are newer ones.

Calcium effects in other age groups are usually evaluated in combination with vitamin D, so data for calcium alone are limited (Newberry et al., 2014).

Response: We have modified the phrase to indicate we were referring to bone health effects and we have also change the verb to the past tense.

Introduction: “Calcium effects on bone health in other age groups are usually evaluated in combination with vitamin D, so data for calcium alone are limited”

57-59 „The benefits of calcium supplementation …” - missing info how often and which regions

The benefits of calcium supplementation seem to be greater in children and adolescents with low calcium intake (Weaver et al., 2016).

Response:

Thank you, but at the moment there is not enough information to recommend distinctively for different world regions or to propose a frequency.

72-73 - sentence „… as there are major acceptability and feasibility concerns …” is unclear. What is „acceptability” and „feasibility”? This sentence should be rewritten or split.

Response: we have modified the paragraph and added references to clarify the phrase.

Introduction: “The 2020 update of the guidelines recommend that women achieve an adequate calcium intake through locally available foods and suggest that food fortification might be an appropriate strategy to fulfil this recommendation, as there are major acceptability and feasibility concerns with recommendation to increase calcium intake using supplements [17,19]. Previous studies have highlighted that the pill burden of calcium supplementation is high, and adherence to medications is lower when multiple doses are required.[19] Also, a small proportion of individuals experience side effects such as constipation, and these side effects may be worse at higher doses. REF Feasibility of implementing this strategy to reach all the population is also a concern as bulk weight of calcium supplements is high resulting in significant logistical cost burdens due to transport, storage and distribution. [20,21] Besides there are economic constrains when supplements need to be paid by the users. These acceptability and feasibility issues are major barriers to scale-up and adoption of calcium supplementation by health systems in low- and middle-income countries. [15,22]”

74-80 – no citation. There is too much information in this paragraph - the authors write both about the effects of an overdose of calcium (this requires expansion) and the economic costs. And at the end of the paragraph there is information that this applies to low- and middle-income countries. Could the side effects not occur in highly developed countries?

Response: We have added citations

Introduction: Previous studies have highlighted that the pill burden of calcium supplementation is high, and adherence to medications is lower when multiple doses are required.[19] Also, a small proportion of individuals experience side effects such as constipation, and these side effects may be worse at higher doses. [20]   Feasibility of implementing this strategy to reach all the population is also a concern as bulk weight of calcium supplements is high resulting in significant logistical cost burdens due to transport, storage and distribution. [20,21] Besides there are economic constrains when supplements need to be paid by the users. These acceptability and feasibility issues are major barriers to scale-up and adoption of calcium supplementation by health systems in low- and middle-income countries. [15,22]

85 – it should be clarified what is „remote populations”

In addition, fortification would reach remote populations or those which lack contact with the health systems increasing the benefits beyond pregnancy, and beyond women  (Cormick and Belizán, 2019).

Response: We have modified the phrase to clarify we refer to populations with less contact with the health system.

Introduction: “In addition, fortification would reach populations with less contact with the health systems increasing the benefits beyond pregnancy, and beyond women” 

115-117 „The only case in which we accepted … no evidence of effect modification” – it is no clear what effect modification is about, this whole sentence is unclear

Response:  We have rephrased the sentence.

Methods: “We only included studies with calcium and vitamin D fortification when the outcome was dietary calcium intake, since there is no evidence that vitamin D interferes calcium intake. Otherwise, we only considered calcium fortification as an intervention.”

128 – in this place authors indicate that language was not limitation, but 1 article was excluded before language (Figure 1) – for what reason this article was excluded?

Response: We did not apply language limitations or publication date restrictions for the search, however we only included studies in English and Spanish we modified the text to specify this.

Methods:

“We did not apply language limitations or publication date restrictions for the search, however we only included studies in English and Spanish.”

155 – Statistical analyses - Did the authors check the possibility of increasing the confidence interval (e.g. up to 97%) in order to narrow the obtained numerical ranges, because for many data there is a very wide range, e.g. lines 276-277 (95% CI 147.6 to 297.4)

Response: we decided to keep the standard CI to show the uncertainty level of the evidence with the planned CI and discuss the imprecision of the results obtained. We used the GRADE approach that also incorporated the assessment of imprecision.

173 - the authors write that 5 articles were subject to narrative synthesis, but does not write about it in the methodology. It is unclear what exactly was narrative synthesis, this term may not be clear and requires explanation

Response: We have clarified that 5 studies were only described as we were not able to include them in the meta-analysis.

Methods: “The 15 RCTs were included in the meta-analysis and 5 studies were only described as they we were not able to include them in the meta-analysis.”

226-232 - the authors refer the consumption of Ca to the official recommendations, but do not cite any of them, e.g. Dietary Reference Intakes for Calcium and Vitamin D, Institute of Medicine US, especially as the recommendations for the elderly differ slightly between countries

Response: Thank you, we have clarified this and included the reference. (INSTITUTE OF MEDICINE. Food and Nutrition Board., 2011)

Discussion: “We found that fortification increased calcium intake in all studied age groups (moderate certainty‑evidence) for children and low certainty-evidence for adults, indicating that fortified foods could be an effective way to help populations attain dietary calcium recommendations.[6]”

422-431 – the relationship between decreased PTH level and higher calcium intake should be better described, because the fact is that increase of Ca serum concentration increases PTH secretion. The authors explain that there may have been an increased incorporation of Ca in the bone, so it should be discussed in the context of vitamin K, which directly participates in this process

Response:  Thank you, we added a parragraph the discuss this topic.

“Calcium balance is hence actively controlled by a large number of factors. The external balance of calcium (the difference between intake and output) is, in effect, determined by the exchange between the skeleton, the intestine and the kidney. These fluxes are controlled by the action of calciotrophic hormones: parathyroid hormone; 1,25-dihydroxycholecalciferol; calcitonin. It is also influenced by other factors such as sex hormones, growth hormones, corticosteroids and a variety of locally-acting hormones. Vitamin K supplementation at doses obtainable in the diet from a 50 g portion of green leafy vegetables, in combination with calcium and vitamin D has been suggested to play a role in the optimization of bone health.”

432-434 - it seems that the bias conclusion should refer to all results obtained and discussed, not just "no adverse effects of fortification"

Response:

Thank you, a paragraph was inserted in the discussion, expanding conclusions about certainty of the evidence in the other outcomes considered.

Discussion: “Except from the outcomes ‘calcium intake’ in adults and postmenopausal women, in which the certainty of evidence was low, for the remaining clinical outcomes considered, it was deemed moderate.”

Because calcium absorption is related to vitamin D and in some studies included to this meta-analysis was also analyzed supplementation with this vitamin, the authors should additionally compare the results from studies where only calcium was used with those where there was calcium + vitamin D (and of course this should be discussed). If this is impossible, there should be information about it, but it should be discussed anyway

If extended the manuscript vitamin D information would exceed the required volume of the manuscript, then you can resign from such extensive data on the intake of Ca, - it is obvious  that the consumption of Ca was higher in the supplemented groups.

Response:

Thank you, our objective was to characterize and isolate the effect of calcium supplementation. Calcium in combination with other nutrients, was outside the scope of our study. We specified in methods that we only included studies with calcium and vitamin D fortification when the outcome was dietary calcium intake, since there is no evidence that vitamin D interferes calcium intake. Otherwise, we only considered Calcium supplementation as an intervention.

For many reference items DOI is missing, although it exist, e.g. item 6, 7, 8, 12, 14, 15

Response: Thank you, we have included the missing DOIs

Reviewer 2 Report

This systematic review is a novel examination of the effect of foods fortified with calcium on health outcomes. To my knowledge there has not been a study like this. However, a few years a go there was a scoping review of fortification that the authors might acknowledge because it shows calcium was the main nutrient in bone studies of food fortification: Whiting SJ, JP Bonjour, W Kohrt, M Warren, M Kraenzlin. (2016) Food fortification for bone health in adulthood: a scoping review. European Journal of Clinical Nutrition. 70(10):1099-1105 . doi: 10.1038/ejcn.2016.42

Specific Comments

1.Abstract: Line 31 what is % sign for? Line 38 probablyis vague considering line 279 shows highly significant dose-response.

2.Methods:Line 130when referring to Appendix A indicate the list of search terms is found there; line 167 what does lustrummean as it is an obscure term.

3.Appendix B please give references for these excluded studies.

4.Results: Line 198-199 and Table1 -region seems too broad a designation; country is important. As the authors have recently written on calcium concerns in LICs, a further documentation of the type of country would be interesting.That the studies were not done in rural areas an LIC country may have a more affluent population in its cities, so rural vs urban would be good to know. These comments can be added to the text and/or the Table. The authors do acknowledge this on line 443.

5.Results line 212: The reference [26] did not report on the type of calcium added to soy milk. This is an important point as researchers(Weaver) has studied two types of fortification and one was better than the other.[See Zhao Y, Martin BR, Weaver CM. Calcium bioavailability of calcium carbonate fortified soy milk is equivalent to cow’s milk in young women. J Nutr 2005;135:237992.] It is common to contact authors for clarification.

6.Results-section 3.4. Lines the economic analysis the two cited studies are [43] and [44] but here are incorrectly numbered 40 and 41. In line 249 incorrectly says USA.

7.Results line 281-288. The changes in weight taken out of context might not have much meaning. If a child gains weight and height, then a change in weight could be from the greater stature. Wouldnt BMI be more meaningful? Of one gains skeleton, there would be an increase in weight yet weightis often perceived as fatness.

8.Discussion line 476 Statement one country has mandatory flour fortification with calcium [49,50]please name that country. Reference 49 is the UK and reference 50 is the USA. Please cite your own review of this topic as the 1998 reference could be misleading a review was conducted in 2013 to keep the fortification as it was.

9.Line 481. Should there be a call for safetyevaluation of calcium fortification for men?

10.Line 496-7. Would you not agree that the UK is an ongoing experimentin calcium fortification?

Author Response

Reviewer 2

Comments and Suggestions for Authors

This systematic review is a novel examination of the effect of foods fortified with calcium on health outcomes. To my knowledge there has not been a study like this. However, a few years a go there was a scoping review of fortification that the authors might acknowledge because it shows calcium was the main nutrient in bone studies of food fortification: Whiting SJ, JP Bonjour, W Kohrt, M Warren, M Kraenzlin. (2016) Food fortification for bone health in adulthood: a scoping review. European Journal of Clinical Nutrition. 70(10):1099-1105 . doi: 10.1038/ejcn.2016.42

Response: Thank you, we have included this reference.

Introduction: “Calcium effects on bone health in other age groups are usually evaluated in combination with other micronutrients specially vitamin D, so data for calcium alone are limited [8] [9].”

Specific Comments

1.Abstract: Line 31 –what is % sign for?

Response: we have modified the sign.

“fortification levels between 96 and 1200 mg per 100 g of food”

Line 38 –“probably” is vague considering line 279 shows highly significant dose-response.

Response: we have deleted probably.

2.Methods:Line 130–when referring to Appendix A indicate the list of search terms is found there;

Response: we have added the information.

Methods: “Appendix A shows the search strategy and the list of search terms”

line 167 –what does “lustrum” mean as it is an obscure term.

Response: Thank you, we have modified the phrase.

Methods: “Due to the scarcity of data we couldn’t perform the analysis for WHO region and continent subgroups.”

  1. Appendix B –please give references for these excluded studies.

Response: Thank you, we have included these references.

  1. Results: Line 198-199 and Table1 -region seems too broad a designation; country is important. As the authors have recently written on calcium concerns in LICs, a further documentation of the type of country would be interesting. That the studies were not done in rural areas an LIC country may have a more affluent population in its cities, so rural vs urban would be good to know. These comments can be added to the text and/or the Table. The authors do acknowledge this on line 443.

Response: Thank you, we agree that country is very important, we have added the study country in table 1.

  1. Results line 212: The reference [26] did not report on the type of calcium added to soy milk. This is an important point as researchers (Weaver) has studied two types of fortification and one was better than the other.[See Zhao Y, Martin BR, Weaver CM. Calcium bioavailability of calcium carbonate fortified soy milk is equivalent to cow’s milk in young women. J Nutr 2005;135:2379–92.] It is common to contact authors for clarification.

Response: Thank you, we agree that the type of salt is very important. Most studies reported the type of salt and we have specified this information, however this article does not describe the type of salt used. We have sent an email to the authors, but so far we have not received a response to this.

  1. Results-section 3.4. Lines the economic analysis the two cited studies are [43] and [44] but here are incorrectly numbered 40 and 41. In line 249 incorrectly says USA.

Response: we have change de country to Germany where the study was performed. The Keller study was performed in the USA.

Methods: “The study of Sandmann et al., from Germany aimed to assess the cost of calcium from a wide variety of sources in diverse cities of Germany, while controlling for seasonal variations [41].”

  1. Results line 281-288. The changes in weight taken out of context might not have much meaning. If a child gains weight and height, then a change in weight could be from the greater stature. Wouldn’t BMI be more meaningful? Of one gains skeleton, there would be an increase in weight yet “weight” is often perceived as fatness.

Response: We agree with the reviewer that, particularly in children BMI is very important, however only one study reported BMI and we were not able to meta‑analyze this information. On the other hand, as studies reported weight and height separately we decided to include the results of the meta‑analysis of these two outcomes.

  1. Discussion line 476 –Statement “one country has mandatory flour fortification with calcium [49,50]”please name that country.

Response: We have added the country

Discussion: “only one country, the UK, has mandatory flour fortification with calcium”

Reference 49 is the UK and reference 50 is the USA. Please cite your own review of this topic as the 1998 reference could be misleading –a review was conducted in 2013 to keep the fortification as it was.

Response: We have added the correct citating to the 2013 public consultation

References

Department for Environment, F. and R.A. Bread and Flour Regulations 1998 A summary of responses to the consultation and Government Reply Available online: https://assets.publishing.service.gov.uk/government/uploads/system/uploads/attachment_data/file/226553/bread-flour-sum-resp-130805.pdf (accessed on Dec 10, 2020).

9.Line 481. Should there be a call for safety evaluation of calcium fortification for men?

Response: Thank you we have added a paragraph to highlight this concept.

“We did not find any study for some age groups such as men alone and adolescents that have evaluated clinical outcomes, therefore further studies are needed to assess the calcium fortification effectiveness and safety for these  populations.”

10.Line 496-7. Would you not agree that the UK is an ongoing experiment in calcium fortification?

Response: We agree that the UK experience could be considered as a natural experiment. Therefore, we changed no information with scarce information.

“There is scarce information on the sustainability of these programmes in the long-term.”

Round 2

Reviewer 1 Report

The phrase (On the other hand, as calcium supplements has been related to constipation, …) about elderly was not added to the manuscript.

Citation 5 relates to UK but the whole paragraph discusses US. Does UK have more recent recommentations (after 1992)?

„as there are major acceptability and feasibility concerns with recommendation to increase calcium intake using supplements” – this sentence is still unclear. What do you suggest by acceptability and feasibility concerns in terms of fortification? It will be easier to sell fortified food just because people generally accept calcium supplements? It looks like mental shortcut without any justification.

Replay to my previus comment "422-431 – the relationship between decreased PTH level and higher calcium intake..." text is OK, but has no citation.

Author Response

Dear Editor,

Thank you for the comments we have incorporated the suggested references in the manuscript.

We hope this fulfils your requirements,

With best regards,

Gabriela.

Academic Editor Notes

In the abstract, the authors state that the largest cost savings (43%) came from prevented hip fractures in older women. This is an extrapolation not supported by present evidence. Please, read a recent report from the US Preventive Services Task Force: Kahwati LC, Weber RP, Pan H, Gourlay M, LeBlanc E, Coker-Schwimmer M, Viswanathan M. Vitamin D, Calcium, or Combined Supplementation for the Primary Prevention of Fractures in Community-Dwelling Adults: Evidence Report and Systematic Review for the US Preventive Services Task Force. JAMA. 2018 Apr 17;319(15):1600-1612. doi: 10.1001/jama.2017.21640. Accordingly, the wording both in the abstract and in the Introduction should be modified. The fact is that there is no effect on fracture risk by use of calcium supplements in community-dwelling elderly women and men and even a higher risk of hip fracture as shown in meta-analysis of RCTs (Bischoff-Ferrari HA, Dawson-Hughes B, Baron JA, Burckhardt P, Li R, Spiegelman D, Specker B, Orav JE, Wong JB, Staehelin HB, O'Reilly E, Kiel DP, Willett WC. Calcium intake and hip fracture risk in men and women: a meta-analysis of prospective cohort studies and randomized controlled trials. Am J Clin Nutr. 2007 Dec;86(6):1780-90. doi: 10.1093/ajcn/86.5.1780. Be specific and mention that calcium supplementation can increase the risk of hip fracture. Before calcium fortification can be generally recommended, the population at risk of calcium deficiency needs to be defined. Otherwise a general recommendation of food fortification can generate more net harm than benefits. 

Response: Our review focus on fortification interventions of calcium alone and in the general population, the reviewer suggests references  on supplementation, however we have included these references in the Abstract and Discussion manuscript as we think they are important.

Abstract: The largest cost savings (43%) reported from calcium fortification programs came from prevented hip fractures in older women from Germany.”

 “Our study highlights that calcium fortification leads to a higher calcium intake, small benefits in children’s height and bone health and also important evidence gaps for other outcomes and populations that could be solved with high quality experimental or quasi-experimental studies in relevant groups, especially as some evidence of calcium supplementation show controversial results on the bone health benefit on older adults.”

Discussion: “We did not find any study for some age groups such as men alone and adolescents that have evaluated clinical outcomes, therefore further studies are needed to assess the calcium fortification effectiveness and safety for these populations. Calcium fortification studies should also confirm the be benefits in older adults as the calcium supplementation effects is controversial.[59,60]

There is a clear hip fracture preventive effect with use of the combination of calcium and vitamin D in institutionalized oldest old individuals who have a low calcium intake combined with a low vitamin D status, results driven largely by the Chapuy study published in NEJM 1993, but these results cannot be extrapolated as a cost saving strategy for the whole population.  Provide the reader with also this information. 

Response: We have added this reference in the introduction.

Introduction:  “The US preventive Task Force (USPTF) recommends calcium supplementation plus vitamin D based on a 2016 meta‑analysis showing a relative risk reduction (RRR) of 15% on the incidence of fractures and a 30% in hip fractures in middle‑aged to older adults (McNellis and Barnes, 2014; Weaver et al., 2016). This is also supported by a study published in 1992 that found that calcium supplementation with vitamin D3  reduces the risk of hip fractures and other nonvertebral fractures among elderly women.(Chapuy et al., 1992)

Not mentioned by the authors, there is a possible transient effect on BMD by increasing calcium intake that is not long-lasting. RCTs has indeed shown that the effect on BMD is most pronounced during the first year of the intervention. My recommendation for the authors is reading of two recent articles: Ian R Reid, Mark J Bolland. Calcium and/or Vitamin D Supplementation for the Prevention of Fragility Fractures: Who Needs It? Nutrients 2020. PMID: 32272593 and Ian R Reid, Sarah M Bristow. Calcium and Bone. Handb Exp Pharmacol 2020. PMID: 31792679. Please describe this issue in the manuscript. 

Response: We modify the conclusion to incorporate these references.

Conclusion: “In addition, studies are needed to explore the effect of higher fortification levels, more appropriate comparison groups in the context of better powered trials and longer intervention periods as there is evidence that there could be a not long-lasting effect.”
